# Discovery of a chemical probe for PRDM9

Abdellah Allali-Hassani[1,10], Magdalena M. Szewczyk[1,10], Danton Ivanochko[1,2,10], Shawna L. Organ[1], Jabez Bok[3], Jessica Sook Yuin Ho[3], Florence P.H. Gay[3], Fengling Li[1], Levi Blazer[1], Mohammad S. Eram[1], Levon Halabelian[1], David Dilworth[1], Genna M. Luciani[1], Evelyne Lima-Fernandes[1], Qin Wu[1], Peter Loppnau[1], Nathan Palmer[3], S. Zakiah A. Talib[3], Peter J. Brown[1], Matthieu Schapira[1,4], Philipp Kaldis[3,5], Ronan C. O'Hagan[6], Ernesto Guccione[3,7,8], Dalia Barsyte-Lovejoy[1,9], Cheryl H. Arrowsmith[1,2], John M. Sanders[6], Solomon D. Kattar[6], D. Jonathan Bennett[6], Benjamin Nicholson[6]* & Masoud Vedadi[1,4]*

PRDM9 is a PR domain containing protein which trimethylates histone 3 on lysine 4 and 36. Its normal expression is restricted to germ cells and attenuation of its activity results in altered meiotic gene transcription, impairment of double-stranded breaks and pairing between homologous chromosomes. There is growing evidence for a role of aberrant expression of PRDM9 in oncogenesis and genome instability. Here we report the discovery of MRK-740, a potent ($IC_{50}$: 80 ± 16 nM), selective and cell-active PRDM9 inhibitor (Chemical Probe). MRK-740 binds in the substrate-binding pocket, with unusually extensive interactions with the cofactor S-adenosylmethionine (SAM), conferring SAM-dependent substrate-competitive inhibition. In cells, MRK-740 specifically and directly inhibits H3K4 methylation at endogenous PRDM9 target loci, whereas the closely related inactive control compound, MRK-740-NC, does not. The discovery of MRK-740 as a chemical probe for the PRDM subfamily of methyltransferases highlights the potential for exploiting SAM in targeting SAM-dependent methyltransferases.

[1] Structural Genomics Consortium, University of Toronto, Toronto, ON M5G 1L7, Canada. [2] Princess Margaret Cancer Centre and Department of Medical Biophysics, University of Toronto, Toronto, ON M5G 2M9, Canada. [3] Institute of Molecular and Cell Biology (IMCB), Agency for Science, Technology and Research (A*STAR), Singapore, Singapore. [4] Department of Pharmacology and Toxicology, University of Toronto, Toronto, ON M5S 1A8, Canada. [5] National University of Singapore (NUS), Department of Biochemistry, 117597 Singapore, Singapore. [6] Merck & Co., Inc., 2000 Galloping Hill Road, Kenilworth, NJ 07033, USA. [7] Department of Oncological Sciences and Tisch Cancer Institute, Icahn School of Medicine at Mount Sinai, New York, NY 10029, USA. [8] Department of Pharmacological Sciences and Mount Sinai Center for Therapeutics Discovery, Icahn School of Medicine at Mount Sinai, New York, NY 10029, USA. [9] Nature Research Center, Vilnius, Akademijos 2, Lithuania. [10] These authors contributed equally: Abdellah Allali-Hassani, Magdalena M. Szewczyk and Danton Ivanochko. *email: benjamin.nicholson@merck.com; m.vedadi@utoronto.ca

PRDM (PRDI-BF1 and RIZ homology domain containing) proteins comprise a distinct branch of known and putative protein lysine methyltransferases (PKMTs) (Supplementary Fig. 1). All PRDMs contain an N-terminal PR domain that shares the canonical SET domain methyltransferase fold, but with only 20–30% amino acid sequence identity[1]. The PRDMs are further distinguished among methyltransferases by a variable number of C-terminal C2H2 zinc-finger motifs with sequence specific DNA binding activity. Humans possess 19 PRDM-coding genes with roles in cellular proliferation and differentiation. Some PRDMs are reported to be tumor suppressors such as PRDM1, 2, 3, 5, 11, 12, and 16, whereas others such as PRDM4, 9, 10, and 14 have been implicated in cancers when their gene expression is dysregulated (reviewed in[2,3]). Methyltransferase activity has been reported for various PRDMs[2]; however, only the activity of PRDM9[4,5], PRDM7[6], and PRDM16[7] have been fully characterized in vitro.

PRDM9 is normally expressed solely in germ cells entering meiotic prophase in female fetal gonads and in postnatal testis[8]. It was first identified as a lysine methyltransferase with activity toward histone 3 lysine 4 (H3K4)[8]. In PRDM9-deficient testis, lower levels of H3K4 trimethylation and altered meiotic gene transcription led to impairment of double-stranded breaks and pairing between homologous chromosomes. These data identified an essential function for PRDM9 in spermatocytes and progression of meiotic prophase[8]. Disruption of PRDM9 in mice results in sterility in both sexes[8]. Allelic variants of PRDM9 zinc fingers have been correlated with variability in genome-wide meiotic recombination sites (hotspots) in humans[9]. Several PRDM9 single nucleotide polymorphisms (SNPs) have been reported for a group of sterile male patients with azoospermia[10].

*PRDM9* has also been identified as a meiosis-specific cancer/testis gene[11], and there is growing evidence that PRDM9 may be involved in oncogenesis and/or cancer evolution. In head and neck squamous cell carcinoma PRDM9 is recurrently mutated[12], while an excess of rare PRDM9 alleles has been reported in aneuploid and infant B-cell precursor acute lymphoblastic leukemia patients[13]. Recent analysis of 1879 cancer samples in 39 different cancer types revealed that PRDM9 is expressed in 20% of tumors even after stringent gene homology correction, and its level of expression in tumors was also significantly higher than in healthy neighboring tissues and in healthy non-germ cell tissue databases[14]. PRDM9 expression correlated with the induction of a meiotic transcriptional program and chromosomal breakpoints at sites of PRDM9-DNA binding[14]. These data support an association between aberrant PRDM9 expression and genomic instability in cancer[14].

The human PRDM9 protein consists of an N-terminal Krüppel associated box (KRAB) domain, a nuclear localization signal, an SSXRD motif, a pre-SET zinc-knuckle motif preceding the PR domain, followed by one post-SET and a distal array of 13 zinc-finger motifs at the C-terminus[15,16]. The KRAB domain functions as an interaction scaffold with CXXC1, PIH1D1, CHAF1A, CEP70, FKBP6, IFT88, and MCRS1 proteins, while the zinc-finger array confers specific genomic localization at speciated DNA sequences[17,18].

PRDM7 is a close orthologue of PRDM9 (Supplementary Fig. 1) and arose from a gene duplication in primates[19]. The amino acid sequences of these two proteins are 97% identical within the PR domain, differing by only three amino acids[6]. However, outside the PR domain, the two proteins have major structural differences including four zinc-finger domains for PRDM7 versus 14 in PRDM9, suggesting differential genomic binding sites. Unlike PRDM9 whose expression is normally restricted to germ cells, PRDM7 is expressed in other tissues[8,19].

Biochemical characterization of PRDM9 catalytic activity with a histone peptide (residues 1–25 of H3) as an in vitro substrate indicated that PRDM9 is the most active histone methyltransferase characterized in vitro, trimethylating H3K4me2 with a $k_{cat}$ value of $18,000 \pm 900\,h^{-1}$[4]. It was also active and capable of mono- and dimethylating H3K4me0 or H3K4me1. Interestingly, PRDM9 is also able to mono-, di-, and trimethylate H3K36 in vitro and in HEK293 cells transfected with PRDM9 plasmid[4]. In addition, Koh-Stenta et al., report PRDM9 activity in vitro with a wider range of substrates[5]. In contrast to PRDM9, PRDM7 is significantly less active ($190\,h^{-1}$) than PRDM9 with H3K4me2 (1–25) histone peptide as a substrate and shows no activity with H3K36 peptides as substrate. Mutation of S357 to tyrosine in PRDM7 restored a pattern of substrate specificity similar to that of PRDM9[6].

Although an increasing number of selective inhibitors for human methyltransferases have been reported in recent years[20], no inhibitors have yet been reported for the PRDM subfamily of enzymes. Here we describe the discovery of MRK-740, a first-in-class chemical probe for PRDM9 that selectively inhibits its methyltransferase activity in biochemical and cellular assays, and the closely related inactive control compound, MRK-740-NC.

## Results

**Discovery of MRK-740.** In order to identify small molecule inhibitors of PRDM9, we used a radioactivity-based methyltransferase assay to screen a library of 7500 compounds, including some of the most diverse compounds in the MSD compound collection, compounds similar to known methyltransferase inhibitors, and candidate compounds from a virtual screening campaign. We identified 39 screening hits, which inhibited PRDM9 with $IC_{50}$ values ranging from 4 to 30 μM at concentrations of SAM and substrate equivalent to their respective $K_m$ values (balanced conditions)[4]. All confirmed hits were subsequently chemically validated by re-purification of the original material or de-novo synthesis. After a few rounds of initial structure-activity-relationship (SAR) generation, we identified small molecules for further studies (Supplementary Fig. 2 and Supplementary Data 1). An extensive follow up SAR campaign led to the identification of MRK-740 as a potent PRDM9 inhibitor with $IC_{50}$ value of $80 \pm 16$ nM (mean ± standard deviation) (Fig. 1a, c, Supplementary Data 2 and 3, and Supplementary Methods). We also synthesized an inactive control compound (MRK-740-NC) by replacing the methyl pyridine moiety of MRK-740 by a phenyl group (Figs. 1b, 1c and Supplementary Data 2 and 3). Due to the high sequence similarity between PRDM9 and PRDM7 in the PR domain (97% identity)[6], we tested inhibitory activity on PRDM7. Interestingly, MRK-740 was much less potent at inhibiting PRDM7 ($IC_{50} = 45 \pm 7$ μM) while MRK-740-NC had no significant PRDM7 inhibition (Fig. 1d).

The potency of MRK-740 was further assessed by orthogonal methods. Surface plasmon resonance (SPR) analysis also confirmed its binding to PRDM9 with a $K_d$ value of $87 \pm 5$ nM in the presence of 350 μM of SAM and $k_{on}$ and $k_{off}$ values of $1.2 \pm 0.1 \times 10^6\,M^{-1}\,s^{-1}$ and $0.1 \pm 0.01\,s^{-1}$, respectively, as determined by kinetic analysis of the SPR data (Figs. 2a and 2b). Binding of MRK-740 to PRDM9 and PRDM7 as well as 11 other PRDM family members was tested using differential scanning fluorimetry (DSF)[21] as described in Methods. In this qualitative assay, the increase in stability of protein upon binding of small molecules is monitored and any stabilization ($\Delta T_m = T_{m+ligand} - T_{m\ apo}$) of greater than 2 °C is considered significant and a confirmation of binding. As expected, of all the PRDMs tested, 250 μM MRK-740 only stabilized PRDM9 and PRDM7 with $\Delta T_m$ values higher than 2 °C (Fig. 2c; Supplementary Fig. 3). Using the enzyme activity

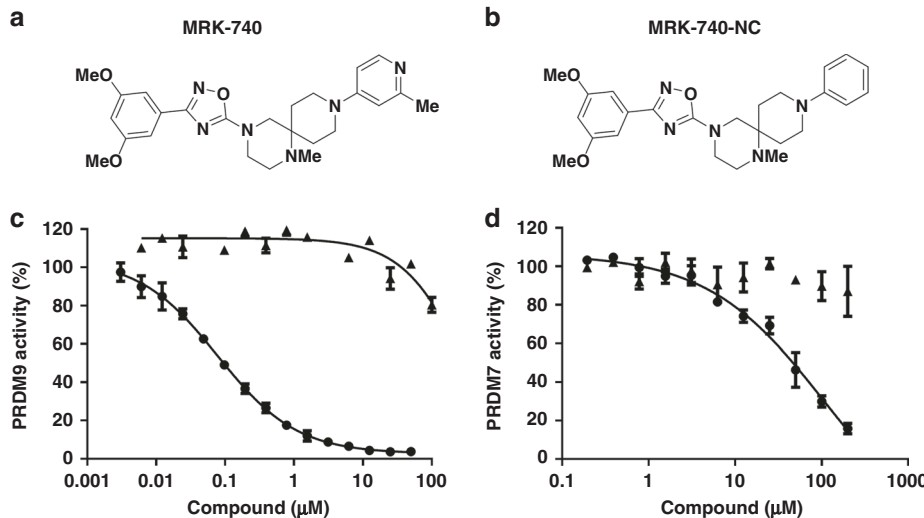

**Fig. 1 PRDM9 inhibition by MRK-740.** Chemical structures of (**a**) MRK-740 and (**b**) MRK-740-NC are presented. Inhibitory effect of (circle) MRK-740 and (triangle) MRK-740-NC on the methyltransferase activity of (**c**) PRDM9 and (**d**) PRDM7 were evaluated. Experiments were performed in triplicate using biotinylated H3 (1–25) peptide as a substrate as described in Methods. Data points are the average ± standard deviation from a minimum of three repeats ($n = 3$). Source data are provided as a Source Data file.

assay, MRK-740 was selective against 32 protein, DNA and RNA methyltransferases (Supplementary Fig. 4) and PRDM7 (Fig. 1d). These data further confirm the selectivity of MRK-740 for PRDM9. MRK-740 was profiled in a customized panel of 108 enzymes and receptors at Eurofins Panlabs Discovery Services, Taiwan. MRK-740 demonstrated significant (>50% at 10 μM) binding only to Adrenergic α$_{2B}$, Histamine H3, Muscarinic M2 and Opiate μ receptors. Follow up functional assessment at the University of North Carolina at Chapel Hill (https://pdspdb.unc.edu/pdspWeb/) for these four GPCRs in agonist and antagonist mode revealed no functional activity on Adrenergic α$_{2B}$, Histamine H3 & Muscarinic M2 receptors, and some potential agonistic activity at the Opiate μ receptor (MOR) that was not concentration-dependent (Supplementary Fig. 5).

We next investigated the mechanism of PRDM9 inhibition by determining the IC$_{50}$ values of MRK-740 at various concentrations of SAM and peptide substrate (Fig. 2d, e). We observed an increase in IC$_{50}$ values as the peptide concentration was increased, indicative of a competitive pattern of PRDM9 inhibition with respect to peptide (H3 1–25) substrate[22] with a $K_i$ value of 65 ± 6 nM (calculated from $Y$ intercept of Fig. 2e). A decrease in IC$_{50}$ values was observed as the SAM concentration was increased, consistent with an uncompetitive pattern with respect to SAM[22]. These data indicate that while MRK-740 competes with the peptide substrate, MRK-740 binding is SAM-dependent and its binding improves as the concentration of SAM is elevated (Fig. 2d, e). However, SAH did not have the same effect on PRDM9 inhibition by MRK-740 (Supplementary Fig. 6).

**Crystal structure of PRDM9 in complex with MRK-740.** To better understand the molecular interaction between PRDM9 and MRK-740 and mechanism of inhibition, we determined the crystal structure of the PRDM9 catalytic domain (residues 195–385) in a ternary complex with SAM and MRK-740 at 2.58 Å resolution (Supplementary Table 1). Clear electron density for MRK-740 was located adjacent to the SAM-binding site and allowed unambiguous placement of the inhibitor molecule (Fig. 3a and Supplementary Fig. 7). An analysis of the intermolecular contacts revealed a network of largely hydrophobic interactions between PRDM9 side chains and MRK-740. Additionally, the pyridine ring of MRK-740 forms a π-stacking

interaction with W356. The pyridine nitrogen is presumed to be protonated due to the basicity of the isolated 4-aminopyridine group (pKa = 9.2) and forms a long-range electrostatic interaction with the sidechain of Asp325 (Fig. 3b). In combination with the methyl group installed on the pyridine and occupying a hydrophobic cavity, this polar contact likely explains the significant difference in PRDM9 inhibition activity observed for MRK-740 and MRK-740-NC, the latter of which contains a phenyl group which would be expected to interact less favorably with the protein. Significant additional interactions were observed between the adenosine moiety of SAM and the two central rings of MRK-740 (Fig. 3b). Collectively, the extensive interaction surface between SAM and MRK-740 explains why we observed SAM-dependent inhibition. None of the three residues differentiating the PRDM9 and PRDM7 PR domains (N289S, W312S, and Y357S) interacts directly with the inhibitor, but Y357 and N289 are within 5 and 7 Å respectively of MRK-740 and may contribute to the specificity of inhibition via second-shell factors such as the stabilization of residues directly interacting with the inhibitor.

To gain insight into the substrate-competitive aspect of MRK-740 inhibition we compared our inhibitor-bound structure to that of mouse PRDM9 in complex with a H3K4me2 substrate peptide and S-adenosyl-homocysteine (SAH) (PDB ID: 4C1Q)[15]. The binding position of the substrate lysine is sterically occluded by both methoxy groups of the MRK-740 benzene ring (Fig. 3c). Furthermore, while both protein structures aligned with a small root-mean-square-deviation (rmsd) of 0.461 Å overall, there was a large difference in the conformation of the substrate recognition helix. In the MRK-740-bound protein, the substrate recognition helix is flipped in the opposite direction, thereby preventing the enzymatically active conformation seen in the mouse structure (Fig. 3d). Taken together, the structural data are consistent with our biochemical results indicating a SAM-dependent, peptide-competitive mechanism of action (MOA) for MRK-740.

Remarkably, our crystal structure revealed that SAM forms an extensive structural component of the MRK-740 binding pocket. To evaluate the uniqueness of this mode of binding, we analyzed the 97 methyltransferase structures in the PDB that had small-molecule inhibitors bound in close proximity to SAM or SAH. For each structure we tallied the number of ligand atoms at the

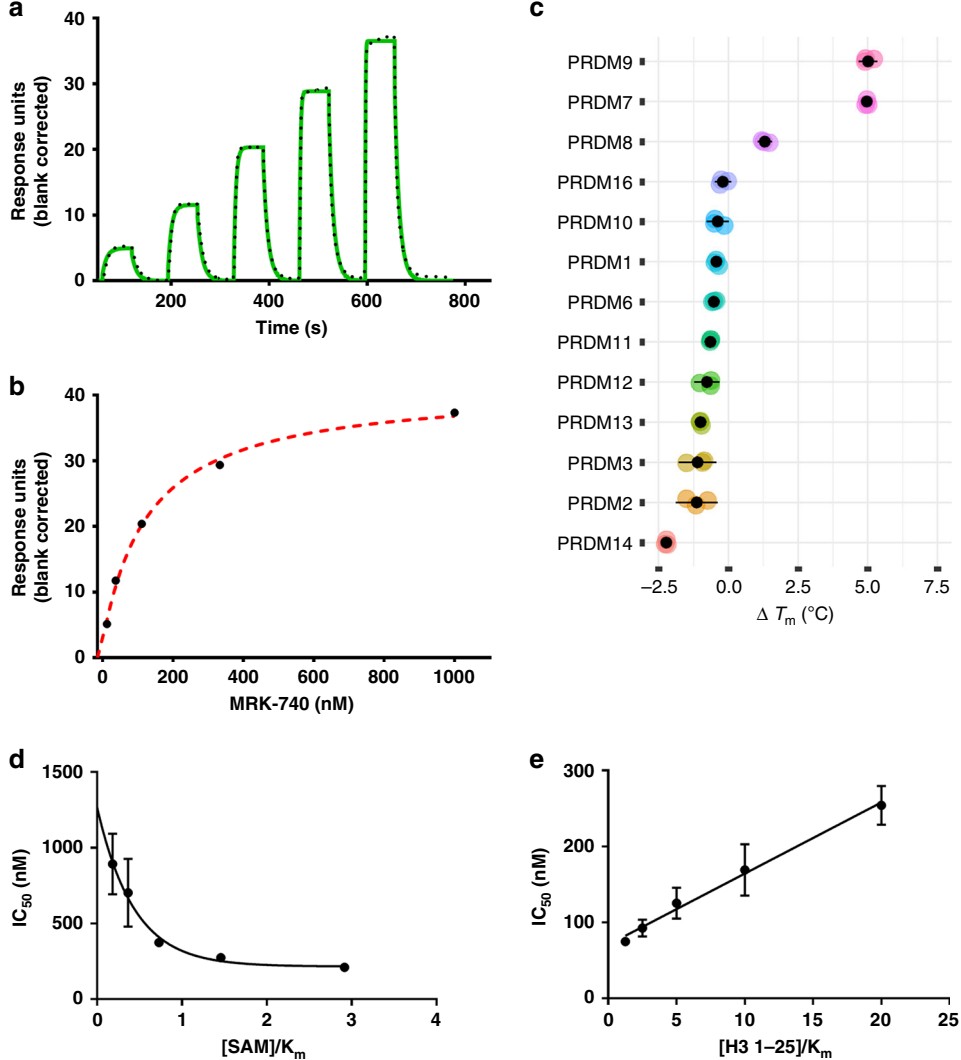

**Fig. 2 Orthogonal confirmation of MRK-740 binding to PRDM9.** Binding affinity of MRK-740 was determined using SPR in the presence of 350 μM SAM ($5xK_{m\,SAM}$, 70 ± 10 μM). **a** A representative sensorgram (solid green) is shown with the kinetic fit (black dots). From kinetic fitting, a $K_d$ value of 87 ± 5 nM, $k_{on}$ of 1.2 ± 0.1 × 10$^{+6}$ M$^{-1}$ S$^{-1}$ and $k_{off}$ of 0.1 ± 0.01 s$^{-1}$ were obtained from quadruplicate experiments ($n = 4$). **b** The steady state response (black circles) obtained from A is shown with the steady state 1:1 binding model fitting (red dashed line). Biotinylated PRDM9 (amino acids 195–415) was immobilized on the flow cell of a SA sensor chip in 1x HBS-EP buffer, yielding 5700 RU. Using the buffer with 0.5% DMSO, 350 μM SAM and single cycle kinetic with 60 s contact time and a dissociation time of 120 s at a flow rate of 75 μL/min. MRK-740 was tested at 1 μM as the highest concentration with dilution factor of 0.33 for five tested concentrations. **c** Binding and selectivity of MRK-740 against PRDM family members were tested by DSF as described in Methods. Only binding to PRDM9 and PRDM7 with $\Delta T_m$ of higher than 2 °C was observed. Mechanism of action of MRK-740 was also evaluated by determining the IC$_{50}$ values (**d**) in the presence of fixed peptide (20 μM) and varying SAM concentrations as well as (**e**) fixed concentration of SAM (350 μM) and varying concentrations of peptide. The data indicate that MRK-740 is a SAM-dependent peptide-competitive inhibitor. All (**c–e**) experiments were performed in triplicate ($n = 3$) and data points are presented as average ± standard deviation. Source data are provided as a Source Data file.

interface (distance < 4 Å) with SAM or SAH, and the number of cofactor atoms at the interface with the ligand (Fig. 4a). The interface between SAM and MRK-740 was by far the most extensive with 27 SAM atoms and 17 ligand atoms involved (Fig. 4a, b). While the PR domain of PRDM9 is similar to the canonical SET domain fold, the interaction between MRK-740 and SAM was a significant outlier when compared to all other SET methyltransferase inhibitors. We observed that other SAM-dependent, peptide-competitive SET inhibitors typically bound within the substrate pocket and interacted primarily with the labile methyl group, as seen for SUV420H1, SETD7 and SMYD2 inhibitors (Fig. 4c–e). In these cases, the adenosyl moiety of SAM that is accessible to MRK-740 is buried within the SAM-binding pocket formed, in part, by the post-SET subdomain of canonical

SET enzymes. In the crystal structures of SETD7 and SMYD2, the lysine mimetic pyrrolidine moiety of PFI-2 and AZ505 (respectively) makes van der Waals interactions with the departing methyl group of SAM, while the phenyl moiety of the ligand (9ZY; PDB ID: 5WBV) obstructs the methyl group of SAM in SUV420H1. Several Rossmann-fold methyltransferases such as HNMT, PRMT5 and COMT had more extensive ligand-cofactor interactions, although still less than PRDM9-MRK-740 (Fig. 4f–h). Quinacrine bound to HNMT makes several van der Waals interactions with the ribose and thioester of SAH (Fig. 4f), while the PRMT5 inhibitor EPZ015666 makes van der Waals interactions as well as a cation–π interaction with the departing methyl group on the positively charged sulfonium of SAM (Fig. 4g).

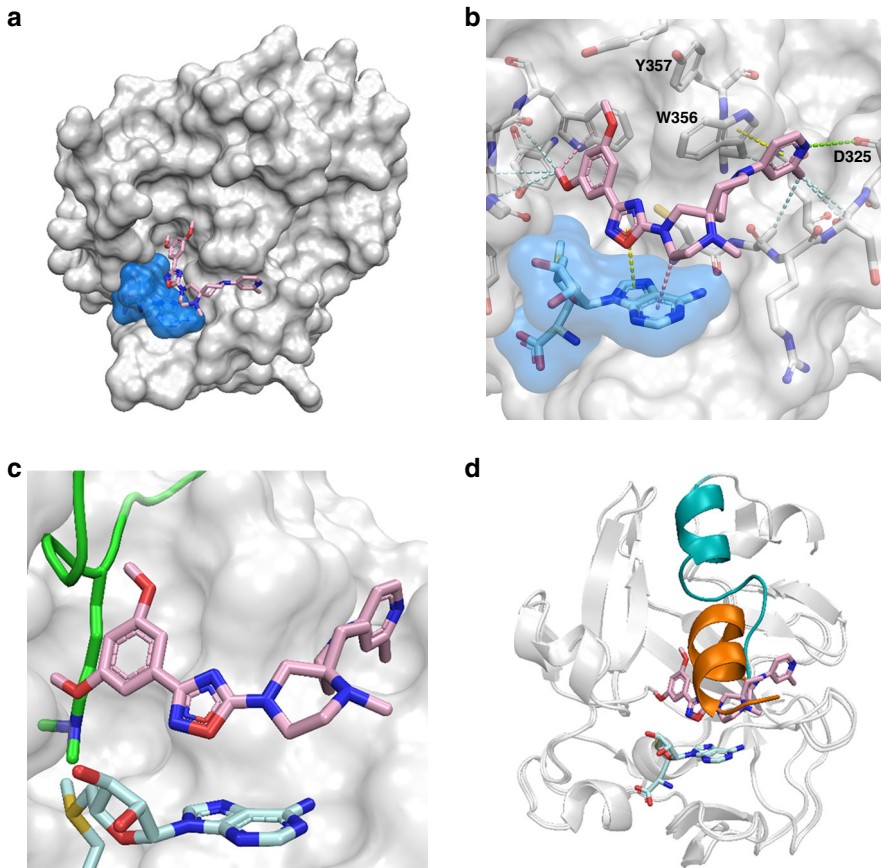

**Fig. 3 Co-crystal structure of MRK-740 bound to PRDM9. a** Surface representation of PRDM9 PR domain (white) bound by SAM (blue) and MRK-740. **b** Intermolecular interactions between MRK-740 and its binding pocket. Hydrophobic interactions (light blue), pi–pi interactions (yellow), CH–pi interactions (pink) and polar interactions (green) are represented by dashes. Structural alignment with the mouse holoenzyme (PDB accession code 4C1Q) showing steric clashes between MRK-740 and (**c**) the substrate lysine residue (green) and (**d**) the post-SET substrate recognition helix when in an active, substrate-bound conformation (orange). The corresponding helix in the inhibited, human structure is shown in (teal).

**WaterMap simulation**. While our structures explain the SAM dependence of MRK-740 binding, the lack of binding in the presence of SAH was not immediately obvious. We reasoned that the void formed by removal of the SAM sulfonium methyl group in the hypothetical PRDM9/SAH/MRK-740 complex could trap and isolate one or more water molecules from bulk solvent, resulting in an unfavorable complex due to the inclusion of water molecules with suboptimal hydrogen bond networks. In order to test this hypothesis, we performed WaterMap simulations[23,24]. Three unfavorable hydration sites with relatively large and positive enthalpies (indicating unsatisfied hydrogen bond networks relative to bulk solvent) were identified in the PRDM9/SAH/ MRK-740 complex (red/brown spheres, Supplementary Fig. 8a). In the absence of MRK-740, water molecules surrounding SAH were energetically less unfavorable, perhaps facilitating dissociation of the reaction product SAH during the catalytic cycle. In comparison, simulation of the PRDM9/SAM/MRK-740 complex did not identify any high-energy hydration sites near the location of the SAM sulfonium group (Supplementary Fig. 8b), suggesting that SAM is able to exclude the binding of energetically unfavorable water molecules through steric means. The exclusion of unfavorable water molecules would result in a complex of increased stability relative to the PRDM9/SAH/MRK-740 complex, consistent with our experimental observations.

**MRK-740 inhibits H3K4 trimethylation by PRDM9 in cells**. In order to demonstrate target engagement in cells, we first used a NanoLuciferase thermal shift assay (NALTSA)[25] as described in methods. In this assay we measured the stability of PRDM9 overexpressed in HEK293 cells after treatment with 10 μM of MRK-740 or MRK-740-NC in 0.1% DMSO for 1 h (Fig. 5). Cells treated with 0.1% DMSO alone were used as a control. PRDM9 showed a $T_m$ of $52 \pm 0.3\,°C$ in the absence of either compound, and was stabilized by MRK-740 ($T_m$ of $54 \pm 0.2\,°C$; $\Delta T_m = 2\,°C$), but not MRK-740-NC ($T_m$ of $51.6 \pm 0.3\,°C$), a clear indication of MRK-740 engaging PRDM9 in cells.

We also overexpressed FLAG-tagged PRDM9 in HEK293T cells together with GFP-tagged histone H3. Ectopic expression of wild-type PRDM9, as reported previously[4], was able to increase both ectopic and endogenous H3K4me3 levels, while no changes were observed with a catalytically dead mutant PRDM9 (Fig. 6a, Supplementary Fig. 9). After 20 h of treatment, MRK-740 reduced PRDM9-dependent trimethylation of ectopic H3K4 ($IC_{50} = 0.8 \pm 0.1\,μM$) (Fig. 6a, b) and endogenous H3K4 (Supplementary Fig. 9) in a concentration-dependent manner, whereas MRK-740-NC did not inhibit H3K4 trimethylation up to concentrations as high as 10 μM (Fig. 6a, b). After 24 h of treatment, neither compound affected HEK293T cell growth at the $IC_{90}$ of MRK-740 (3 μM) but some toxicity was observed at 10 μM (Fig. 6c). Following 4 days of treatment, cytotoxicity was observed with 10 μM MRK-740 and MRK-740-NC, but not 3 μM MRK-740 or MRK-740-NC in HEK293T cells (Supplementary Fig. 10). To further investigate the inhibitory activity of MRK-740, we tested MRK-740 in a second cell line. Importantly, MRK-740 was an equipotent inhibitor of H3K4 methylation in MCF7

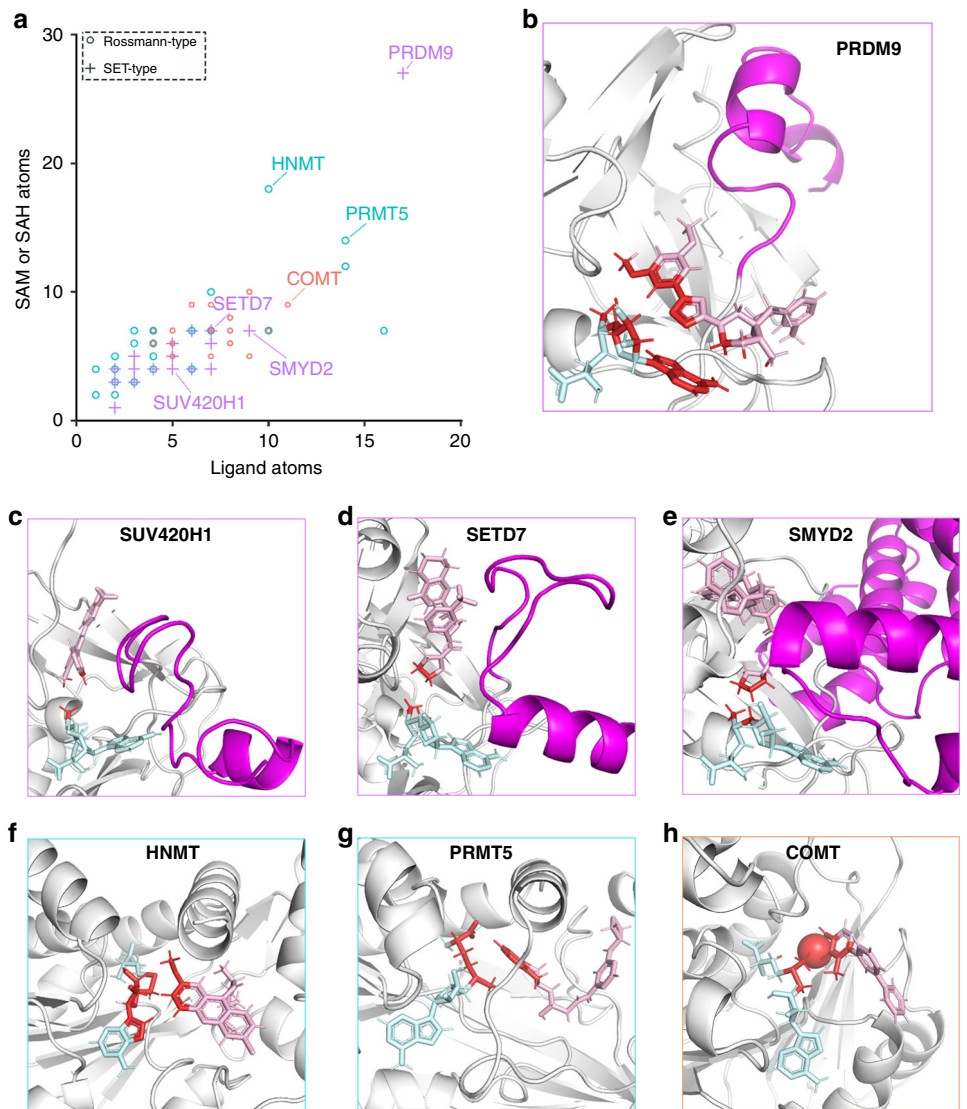

**Fig. 4 MRK-740 binding is unique among methyltransferase inhibitors. a** Atoms tallied at the interface (distance < 4 Å) between SAM or SAH and small molecule inhibitors. Rossmann-type methyltransferases targeting nitrogen or oxygen are indicated in blue or orange, respectively. The tallied atoms (red) at the interface between SAM or SAH (teal) and small molecule inhibitors (pink) are shown for (**b**) PRDM9 with MRK-470, (**c**) SUV420H1 with 9ZY, (**d**) SETD7 with (R)-PFI-2, (**e**) SMYD2 with AZ506, (**f**) HNMT with Quinacrine, (**g**) PRMT5 with EPZ015666, and (**h**) COMT with a $Mg^{2+}$ cation and 43J. The post-SET helices are indicated in purple. PDB accession codes: 6NM4, 5WBV, 4JLG, 5KJN, 1JQE, 4 × 61 and 4XUE, respectively. Data are from Supplementary Table 2.

cells (Supplementary Fig. 11). Furthermore, 10 μM MRK-740 exhibited minimal impact on MCF7 cell viability in a 5-day proliferation assay (Supplementary Fig. 10). These results indicate that MRK-740 is a potent and cell-active PRDM9 inhibitor that together with the negative control compound MRK-740-NC, can serve as a useful tool for studying the physiological and pathological functions of PRDM9.

**MRK-740 inhibits PRDM9 activity on chromatin.** To confirm the ability of MRK-740 to inhibit the methylation of PRDM9 on the more physiologically relevant substrate of endogenous chromatin, we transfected HEK-293T cells with wild-type (PRDM9[WT]) or catalytically dead PRDM9 (PRDM9[Mut]) before treating these cells with various concentrations of MRK-740 or MRK-740-NC for 48 h. ChIP-qPCR assays were then conducted on known PRDM9 binding loci[26]. Ectopic expression of PRDM9[WT] increased trimethylation of H3K4 (H3K4me3) at previously described PRDM9-specific intragenic and intergenic

regions[26] (Fig. 7; Supplementary Fig. 12). In contrast, at transcription start site (TSS) regions, where H3K4me3 deposition may be regulated by multiple pathways, independent of PRDM9, we interestingly observed a decrease in H3K4me3 deposition following PRDM9[WT] overexpression. However, H3K4me3 levels were not further decreased upon MRK-740 treatment, indicating that the PRDM9[WT] overexpression mediated decrease in H3K4me3 was likely to be independent of PRDM9 catalytic activity. No further increase in H3K4me3 levels was observed at the TSSs, intragenic or intergenic regions upon PRDM9[Mut] expression (Supplementary Fig. 12). The observed increases in H3K4me3 levels at intragenic and intergenic regions are therefore due directly to PRDM9 PR domain-dependent methylation activity, rather than through recruitment of other H3K4 methyltransferases or other indirect mechanisms.

Following exposure to MRK-740, increases in H3K4me3 levels were abrogated in a concentration-dependent manner in three out of four PRDM9-dependent loci (Fig. 7, Supplementary

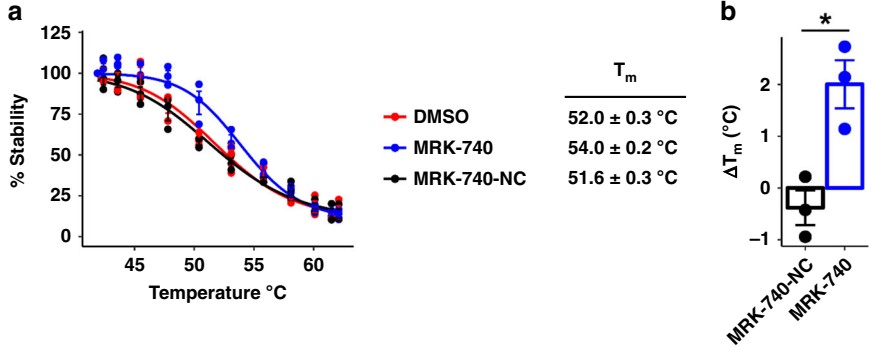

**Fig. 5 MRK-740 engages PRMD9 in a cell-based thermal stability assay. a** Effect of MRK-740 (10 μM) or MRK-740-NC control compound (10 μM) on the thermal stability of NanoLuc-tagged PRDM9 (residues 135–285) in cells. Shown is the mean ± SEM of three biological replicates. **b** $T_m$-shift ($\Delta T_m$) of NanoLuc-tagged PRDM9 in response to MRK-740 (10 μM) or MRK-740-NC control compound (10 μM). Shown is the mean ± SEM of three biological replicates. Boxes indicate the mean values. *P*-values calculated using an unpaired Welch's two-tailed *t*-test, \**p*-value = 0.017. Source data are provided as a Source Data file.

Fig. 12). No significant reduction in H3K4me3 levels was observed at the tested TSS sites upon MRK-740 treatment, even though they exhibit high levels of (PRDM9-independent) H3K4me3. In contrast, incubating transfected cells with various concentrations of MRK-740-NC had a negligible impact on PRDM9-mediated H3K4me3 levels on PRDM9-specific intragenic or intergenic peaks. In summary, these results show that MRK-740 specifically inhibits PRDM9-dependent methylation on endogenous chromatin, but does not affect H3K4 methylation activity from non-PRDM9 sources. Furthermore, as expected, MRK-740-NC does not inhibit PRDM9 activity.

**MRK-740 does not affect proliferation of cancer cell lines**. To determine if MRK-740 impacted proliferation of cancer cell lines, we tested three cell lines with detectable *Prdm9* expression (FPKM ≥ 1) and three with negligible expression (FPKM < 1) as defined by Cancer Cell Line Encyclopedia (https://portals.broadinstitute.org/ccle). The cell lines that express *Prdm9* included the multiple myeloma cell line SKMM2 as well as the two breast cancer cell lines CAL851 and HCC1806. The three matched control cell lines were RPMI-8226, MDA-MB-436 and HCC1143. Treatment with MRK-740 and MRK-740-NC up to 30 μM did not have any significant effect on proliferation of any of the cell lines tested indicating that proliferation of the *Prdm9* expressing cell lines is not PRDM9 dependent (Supplementary Fig. 13).

**MRK-740 treated spermatocytes show meiotic defects**. In order to validate the biological activity of MRK-740 in an endogenous context, we investigated whether administration of this chemical probe in vivo affects meiotic integrity in male germ cells. The major phenotype of *Prdm9*⁻/⁻ mice is an arrest of meiosis during the first meiotic division resulting in infertility. Here, Prdm9 is required to generate an epigenetic profile on chromatin which specifies where meiotic recombination should take place. Recombination is a prerequisite for the pairing of chromosomal homologs. As such, the loss of *Prdm9* leads to failed or non-homologous pairing which arrests meiosis and leads to germ cell death[8]. To bypass the difficulties of drug administration to meiotic cells posed by the blood-testis barrier, we utilized a microinjection approach where compounds were delivered directly to the testis (Supplementary Methods). We treated spermatocytes with MRK-740-NC (Supplementary Fig. 14a) and MRK-740 (Supplementary Fig. 14b, d, e) as described in the supplementary methods. MRK-740 treated spermatocytes showed meiotic defects including non-homologous synapsis or failed

pairing of chromosomal homologs during the pachytene substage of meiotic prophase I (Supplementary Fig. 14b, d, e) that were similar to the defects observed in age matched *Prdm9*⁻/⁻ mice (Supplementary Fig. 14c, f, g). However, MRK-740-NC showed no overt meiotic defects and did not affect pairing of chromosomal homologs during the same stage (Supplementary Fig. 14a). Successful pairing was determined by the co-localization of the synaptonemal complex proteins Sycp3 (magenta staining) and Sycp1 (green staining) (Supplementary Fig. 14).

## Discussion

SAM-dependent methyltransferases perform both essential and pathologic functions in health and disease and are an important class of drug targets[27,28]. Discovery of chemical probes for methyltransferases has enabled considerable advances in our understanding of the roles protein methyltransferases play in epigenetics[20] and diseases such as cancer[29,30]. While small molecule chemical probes and drugs are available for SET domain protein lysine methyltransferases and Rossmann-fold arginine methyltransferases, there are no reported inhibitors or ligands for any of the 19 PRDM proteins.

Here, we report the discovery of MRK-740, a potent, selective and cell-active substrate-competitive PRDM9 inhibitor. A first-in-class PRDM inhibitor, MRK-740 binds to PRDM9 in a SAM-dependent manner with an unusually large SAM interaction interface, exploiting the aromatic system of the adenosyl group of SAM. This mode of binding differs from other SET and non-SET methyltransferase inhibitors and relies on the displacement of the post-SET region from its active conformation. Indeed, the adenosyl group of SAM is generally obstructed by the post-SET domain, and therefore cannot interact with inhibitors[31]. The structural diversity of the post-SET regions observed in PRDMs suggests that this class of methyltransferases may be more amenable to the extended cofactor-inhibitor interactions observed with MRK-740. To date, all human PRDM structures have been solved in the apo state and understanding how these PR domains bind SAM would help rationalize the specificity of MRK-740 and contribute to the development of additional PRDM inhibitors.

Development of SAM-competitive protein methyltransferase inhibitors is challenged with finding cell-penetrant molecules that also bind in the polar SAM-binding pocket[20,31,32]. In contrast, substrate-competitive inhibition is a more frequently employed strategy, which can also depend on interactions with either SAM or SAH. The extensive interaction surface seen between MRK-740 and SAM lies at one extreme of a continuum of inhibitor-cofactor interactions (Fig. 4a). These findings suggest that the rational

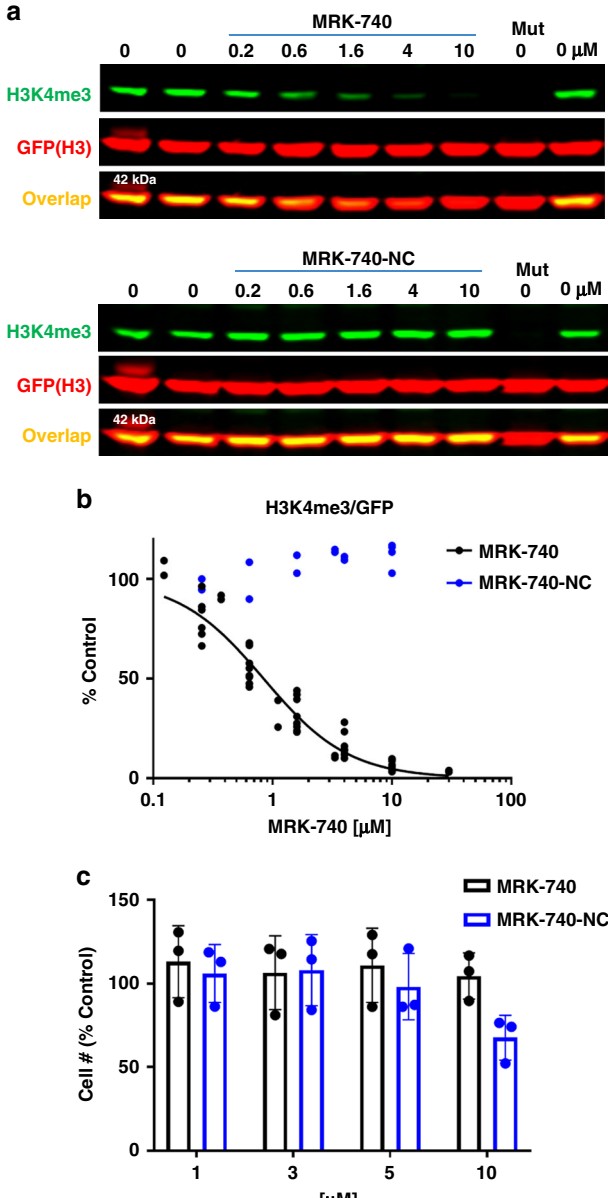

**Fig. 6 MRK-740 inhibits PRDM9 activity in cells. a** MRK-740 but not MRK740-NC decreases PRDM9-dependent K4 trimethylation of exogenous histone H3. HEK293T cells were co-transfected with H3-GFP and PRDM9-FLAG and treated with compounds at indicated concentrations for 20 h. Mut denotes PRDM9 catalytic mutant (Y357S). First and last lanes are controls at 0 μM of compound. **b** The graph represents non-linear fit of H3K4me3 fluorescence intensities normalized to intensities of GFP (MRK-740 $n = 10$, 4 separate experiments, MRK-740-NC $n = 4$, 2 separate experiments). **c** MRK-740 and MRK-740-NC do not affect HEK293T cell growth up to 10 μM. Cells were treated with compounds for 1 day. Cell number was measured using IncuCyte™ ZOOM live cell imaging device. The results are mean ± sd, $n = 3$. Source data are provided as a Source Data file.

design of substrate-competitive small molecule inhibitors could be enhanced by considering the structural interface provided by the bound cofactor, and by understanding subclass-relevant structural dynamics. A question remains as to how selectivity can be achieved using this strategy. SAM adopts a bent conformation in SET and PR enzymes and an extended conformation in Rossmann-fold enzymes[32]. Further selectivity may be

accomplished by exploiting subtle variations in cofactor orientation and in the case of PRDM enzymes, the structural variability of the post-SET regions. As such, our findings of this binding mode are significant for the discovery and design of inhibitors for other SAM-utilizing enzymes.

By using MRK-740, we directly inhibited PRDM9 catalytic activity on chromatin, reducing H3K4me3 levels at intragenic and intergenic target sites. Interestingly, our data also confirm previous observations linking the overexpression of PRDM9 to altered H3K4 methylation levels at the TSS of genes[26]. Such sites typically have low recombination rates, suggesting that PRDM9 could have a role beyond DNA recombination and repair. In addition, mis-regulation of PRDM9 has recently been implicated in cancers, prompting the need for tool compounds to investigate these disease mechanisms[11,14,26]. However, our data did not reveal any significant effect of MRK-740 on proliferation of several cancer cell lines we tested, indicating that at least for the cell lines tested their proliferation was not PRDM9 dependent. To conclude, MRK-740 in combination with its negative control (MRK-740-NC) will provide an excellent set of tool compounds to enable further investigation of the novel physiological and pathological roles of PRDM9.

## Methods

**In vitro assays.** Methyltransferase activity assays for PRDM9 (amino acids: 195–415) were performed by monitoring the incorporation of tritium-labeled methyl group from $^3$H-SAM to H3 (1–25) peptide using scintillation proximity assay (SPA). The enzymatic reactions were performed at 23 °C with 30 min incubation in 20 mM Tris-HCl (pH 7.5), 5 mM DTT, 0.01% Triton X-100 and 1% DMSO containing 71 μM of SAM (5 μM of $^3$H-SAM), 4 μM of biotinylated H3 (1–25) peptide and 5 nM of PRDM9. For PRDM7, reactions were performed at 23 °C with 90 min incubation in 20 mM BTP pH 9.0, 5 mM DTT, 0.01% Triton X-100 and 1% DMSO containing 900 μM of SAM (5 μM of $^3$H-SAM), 4 μM of biotinylated H3 (1–25) peptide and 100 nM of enzyme. Compounds were titrated from 3 nM to 50 μM and from 200 nM to 200 μM for PRDM9 and PRDM7, respectively. To stop the reactions, 10 μL of 7.5 M guanidine hydrochloride was added to 10 μL of reaction mixture, followed by 60 μL of buffer (20 mM Tris, pH 8.0), mixed, and transferred to a 384-well streptavidin coated Flash-plate. After mixing, the mixtures in Flash-plate were incubated for 2 h, and the CPM counts were measured using TopCount plate reader. All enzymatic reactions were performed in triplicate, and IC$_{50}$ values were determined by fitting the data to Four Parameter Logistic equation using the GraphPad Prism software.

PRDM selectivity experiments were carried out by determining the effect of 250 μM of MRK-740 on the thermal stability of PRDM9 and other PRDMs. DSF measurements were performed as previously described[21] using a Light Cycler 480 II instrument from Roche Applied Science. All proteins tested in these selectivity experiments were used at final concentration of 0.02 mg mL$^{-1}$ in a 0.1 M HEPES, pH 7.5 and 150 mM NaCl buffer in the presence of 2 mM SAM. Sypro Orange was purchased from Invitrogen as a 5000 × stock solution (concentration not specified by vendor) and was diluted 1:1000 to yield a 5 × working concentration. The temperature scan curves were fitted to a Boltzmann sigmoid function, and the $T_m$ values were obtained from the midpoint of the transition. Titration experiments by DSF were performed by varying MRK-740 concentrations from 500 nM to 500 μM in the presence or absence of 2 mM SAM.

Selectivity against methyltransferases were determined as described previously[29]. Details of the assays are described in the supplementary methods.

**MOA determination.** To study the MOA of MRK-740, IC$_{50}$ values were determined at either 20 μM fixed concentration of biotinylated H3 1–25 peptide and varying concentrations of SAM from 22 to 350 μM (1.4% $^3$H-SAM) or at 350 μM fixed concentration of SAM and varying concentrations of biotinylated H3 1–25 peptide from 1.25 to 20 μM. Reactions (in quadruplicates) were initiated by adding SAM and followed by measuring the methyltransferase activity using SPA assay. The determined IC$_{50}$ values were fitted against the corresponding substrate concentrations. An increase in IC$_{50}$ values when concentration of peptide was increased indicated a peptide-competitive pattern of inhibition. On the other hand, a decrease in IC$_{50}$ values as the concentration of SAM was increased indicated an uncompetitive pattern of inhibition with SAM indicating that SAM is required for binding.

**Surface plasmon resonance (SPR).** Binding studies by SPR were performed using a Biacore T200 (GE Health Sciences Inc.) at 20 °C. Biotinylated PRDM9 was captured onto a flow cell of a streptavidin-conjugated SA chip at approximately 5000 response units (RU) according to manufacturer's protocol while another flow

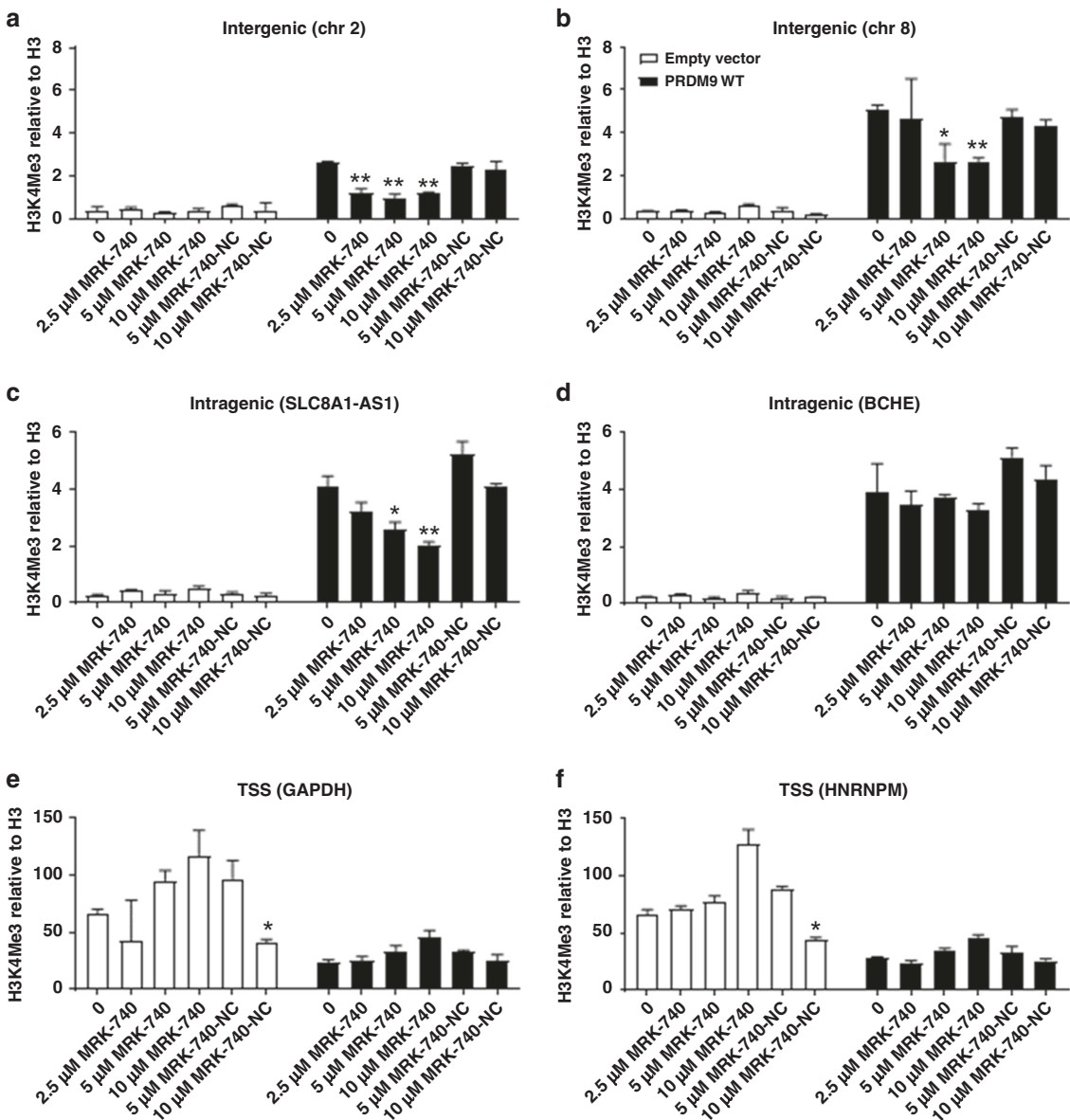

**Fig. 7 Concentration-dependent inhibition of PRDM9-dependent H3K4 trimethylation using MRK-740.** ChIP qPCR analyses of H3K4me3 methylation levels of known PRDM9-bound loci and control loci. Cells transfected with empty vector control, or a vector overexpressing wild-type PRDM9 were treated with DMSO, increasing concentrations of MRK-740 or MRK-740-NC. Concentrations used (μM) are indicated below each plot. (**a–d**) represent reported loci of PRDM9 methylation[26]; (**e, f**) are transcriptional start sites (TSSs) which are not known loci of PRDM9 methylation. Data are normalized to total H3 and are presented as the mean ± upper and lower limits from two replicates. Representative plots of two independent experiments are shown in this figure. The coordinates and the genomic features associated with the assayed loci are indicated above each plot. Left Tailed Student's t-tests were performed by comparing DMSO (0 μM) treated cells to compound treated cells that have been transfected with the same plasmids. *p-value < 0.05 and **p-value < 0.01. Note that here we show a representative of two independent experiments (mean ± s.d of technical replicates) for our ChIP-qPCR experiments. Data obtained using this technique are inherently noisy due to stochastic differences in gene activity/histone positioning. Data are thus typically presented as representative rather than averaged. Source data are provided as a Source Data file.

cell was left empty for reference subtraction. MRK-740 was dissolved in DMSO (50 mM Stock) and diluted to 220 μM before serial dilutions were prepared in DMSO (dilution factor of 0.33 was used to yield 5 concentrations). For SPR analysis the serially titrated compound was diluted 1:200 in HBS–EP (20 mM HEPES pH 7.4, 150 mM NaCl, 3 mM EDTA, 0.05% Tween-20) and 250 μM SAM giving a final concentration of 0.5% DMSO. Kinetic determination experiments were performed using multicycle kinetics with an on time of 60 s, off time of 120 s at a flow rate of 75 μL min$^{-1}$.

**Crystallization and structure determination**. Purified PRDM9 (residues 195–385) was concentrated to 10.9 mg mL$^{-1}$ and incubated for 1 h at 4 °C with 2.5 mM SAM and 2.5 mM MRK-740. Ligand-bound protein crystals were obtained from a 1:1 mixture of protein and crystallization buffer after 3 days by sitting-drop

vapor-diffusion method at 18 °C using 0.1 M BisTris pH6.0, 0.2 M NH4OAc and 24.5% PEG3350 as the crystallization buffer. Single crystals were flash frozen in liquid nitrogen by harvesting them in a cryoprotectant solution using crystallization buffer supplemented with 20% (v/v) glycerol. The dataset was collected at the Advanced Photon Source beamline 24-ID-E with a wavelength of 0.979180 Å from a ADSC Quantum 315 CCD detector at 100 K. All data were processed and scaled with XDS[33] and Aimless[34]. The structure was solved by molecular replacement using a trimmed structure of the human PRDM9 apo-enzyme (PDB accession code 4IJD, chain A, residues 201–354) as the search model with the program PhaserMR[35]. The stereochemical restraints for MRK-740 were generated using the program JLigand v1.0.40[36]. The structural models were refined using REFMAC5[37] and manually checked with COOT. The Ramachandran values were 96.5, 3.0 and 0.5% for favored, allowed and outliers, respectively. Images were generated using PyMOL (The PyMOL Molecular Graphics System, v2.2.0,

Schrödinger, LLC.). Data collection and refinement statistics are shown in Supplementary Table 1.

**WaterMap simulations**. Schrodinger's 2019-2 suite (Schrödinger Release 2019-2, Schrödinger, LLC, New York, NY, 2019) was used for the preparation, simulation, and analysis of WaterMap calculations. The PRDM9/SAM/MRK-740 crystal structure was readied for simulation using Schrodinger's Protein Preparation Wizard utilizing default parameters. The PRDM9/SAH/MRK-740 complex was formed from the prepared PRDM9/SAM/MRK-740 complex via deletion of the SAM sulfonium methyl group and the system was otherwise unchanged from the fully prepared PRDM9/SAM/MRK-740 complex. WaterMap simulations were carried out using default parameters for both complexes.

**Methyltransferase inhibitor survey**. To quantify the interaction between small-molecule inhibitors and SAM or SAH, atoms at their interface (distance < 4 Å) were tallied. Briefly, all SAM or SAH containing protein structures ($n = 1311$) were analyzed from the Protein Data Bank (date of acquisition: 2019-06-01). An in-house Python3 script was used in PyMOL (The PyMOL Molecular Graphics System, v2.2.0, Schrödinger, LLC.) to model hydrogens onto all structures and generate tallies of any ligand atom within 4.0 Å of SAM or SAH and *vice versa*. Next, ligands lacking ring structures or identified as ions or crystallization buffer components were removed and the resulting list of ligands was manually curated to identify the 97 unique inhibitor–SAM/SAH pairs (Supplementary Table 2; Supplementary Data 4).

**PRDM9 cellular assay**. HEK293T (kind gift from Sam Benchimol, York University), or MCF7 (ATCC® HTB-22™), cells were plated (12 well plates) and next day co-transfected with 0.1 (HEK293T) or 0.2 (MCF7) μg of GFP-tagged histone H3 and with 0.1 (HEK293T) or 0.2 (MCF7) μg of Flag-tagged PRDM9/PRDM9 catalytic mutant (Y357S) and 0.8 or 0.6 μg of empty vector using JetPRIME® transfection reagent, following manufacturer instructions. After 4 h media was removed and cells were treated with compounds for 20 h. Next day, cells were lysed in lysis buffer (20 mM Tris-HCl pH 8, 150 mM NaCl, 1 mM EDTA, 10 mM MgCl₂, 0.5% TritonX-100, 12.5 U mL⁻¹ benzonase (Sigma), complete EDTA-free protease inhibitor cocktail (Roche)). After 2 min incubation at RT, SDS was added to final 1% concentration. Cell lysates were analyzed in western blot for GFP and H3K4me3 levels. All cell lines were mycoplasma negative, as determined by MycoAlert™ Mycoplasma Detection Kit (Lonza).

Total cell lysates were resolved in 4–12% Bis-Tris Protein Gels (Invitrogen) and transferred in for 1.5 h (80 V) onto PVDF membrane (Millipore) in Tris-Glycine transfer buffer containing 20% MeOH and 0.05% SDS. Blots were blocked for 1 h in blocking buffer (5% milk in PBS) and incubated with primary antibodies anti-GFP (1:3000, Clontech # 632381), anti-H3K4me3 (1:1000, Millipore, #04-745), anti-histone H3 (1:2000, #ab10799, Abcam), anti-Flag (1:5000, #F4799, Sigma) in blocking buffer (5% BSA in 0.1% Tween 20 PBS) overnight at 4 °C. After five washes with 0.1% Tween 20 PBS the blots were incubated with goat-anti rabbit (IR800) and donkey anti-mouse (IR 680) antibodies (1:5000) in Odyssey Blocking Buffer (LiCor) for 1 h at RT and washed five times with 0.1% Tween 20 PBS. The signal was read on an Odyssey scanner (LiCor) at 800 nm and 700 nm and analyzed with Image Studio Ver.5.2 software.

HEK293T or MCF7 cells were seeded on 96-well plates and the following day treated with compounds for indicated time. Cell number was determined with Vybrant® DyeCycle™ Green using IncuCyte™ ZOOM live cell imaging device (Essen Bioscience) and analyzed with IncuCyte™ ZOOM (2015A) software.

**NanoLuciferase thermal shift assay (NALTSA)**. NALTSA assays were performed as previously described[25]. Briefly, HEK293T cells were plated in 6-well plates at a density of $1.5 \times 10^5$ cells/mL. The following day, cells were transfected with 2 μg of C-terminally NanoLuc®-tagged PRDM9 (residues 135–285) with X-tremeGene HP DNA Transfection Reagent (Roche). Cells were incubated for 24 h, trypsinized and resuspended at $1.5 \times 10^5$ cells/mL in 1x PBS supplemented with protease inhibitors. Cell suspensions were incubated at 37 °C with 10 μM compound or 0.1% DMSO alone for 1 h. Sixty microliters of cells was dispensed per well of a 96-well PCR plate followed by heating across a temperature gradient for 3 min on a Mastercycler® thermal cycler (Eppendorf). After heating, samples were transferred to a 96-well white plastic plate (Corning) and Glo® Substrate (Promega) was added to each well at 1:1000 final dilution. NanoLuciferase signal was read at 450 nm (filter: 450 nm/BP 80 nm) on a CLARIOstar microplate reader (Mandel).

**ChIP-PCR assay**. HEK293T cells were transfected with 37.5 μg empty pcDNA3.1 (+) vector (Invitrogen), or the same vector expressing either wild-type PRDM9 (PRDM9^WT) or a catalytically dead PRDM9 mutant (PRDM9^Mut) and 75 μL of Lipofectamine 2000 (Invitrogen) in 6 mL of Opti-MEM (GIBCO) for 48 h. Concurrently the cells were treated with 0.1% (v/v) DMSO, MRK-740, or MRK-740-NC at the indicated concentrations for 48 h. ChIP assays were then performed as previously described[38]. Briefly, cells were fixed in 1% formaldehyde and the fixation reaction was quenched in a final concentration of 0.1 M glycine. Cell pellets were first lysed in LB1 (50 mM HEPES-KOH pH 7.5, 140 mM NaCl,

1 mM EDTA, 10% Glycerol, 0.5% Igepal, 0.25% TritonX), washed once in LB2 Buffer (10 mM Tris-HCl pH 8, 200 mM NaCl, 1 mM EDTA, 0.5 mM EGTA) and finally sonicated in LB3 buffer (10 mM Tris-HCl pH 8, 100 mM NaCl, 1 mM EDTA, 0.5 mM EGTA, 0.1% Sodium deoxycholate, 0.5% N-laurosylsarcosine) using a bioruptor (Diagenode). Thereafter, 50 μg of chromatin was subjected to ChIP using the following antibodies purchased from Abcam: α-H3K4me3 (1:200, ab8580, lot:GR3207339-1) and α-H3 (1:250, ab1791, lot:GR300978-4). After extensive washing in RIPA wash buffer (50 mM HEPES-KOH pH 7.6, 100 mM LiCl, 1 mM EDTA, 1% Igepal, 0.7% Sodium Deoxycholate), DNA was eluted from beads. The eluted DNA was treated with RNase A and Proteinase K and purified using the QIAGEN PCR purification kit. qPCR was then carried out targeting the indicated loci (Supplementary Table 3). Upper and lower limits were defined by $|2^{-(dCq \pm s)} - 2^{-(\Delta Cq)}|$, where $s = \sqrt{((\text{Standard deviation of Input } Cq)^2 + (\text{Standard deviation of Target } Cq)^2)}$.

**Reporting summary**. Further information on research design is available in the Nature Research Reporting Summary linked to this article.

## Data availability

The refined coordinates and data for the co-crystal structure was deposited in the PDB with the accession code 6NM4[https://www.rcsb.org/structure/6NM4]. The source data underlying Figs. 1c, d, 2a–e, 5a, b, 6a–c, 7a–f, and Supplementary Figs. 3, 4, 5, 9a, b, 10, 11a, b, 12a–f, 13a, b are provided as the Source Data file. Any other data are available from the authors.

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

## Acknowledgements

The SGC is a registered charity (number 1097737) that receives funds from AbbVie, Bayer Pharma AG, Boehringer Ingelheim, Canada Foundation for Innovation, Eshelman Institute for Innovation, Genome Canada through Ontario Genomics Institute [OGI-055], Innovative Medicines Initiative (EU/EFPIA) [ULTRA-DD grant no. 115766], Janssen, Merck KGaA, Darmstadt, Germany, MSD, Novartis Pharma AG, Ontario Ministry of Research, Innovation and Science (MRIS), Pfizer, São Paulo Research Foundation-FAPESP, Takeda, and Wellcome [106169/ZZ14/Z]. This work was supported by funding from the Natural Sciences and Engineering Research Council (PGSD3) to DI. This research used resources of the Advanced Photon Source, a U.S. Department of Energy (DOE) Office of Science User Facility operated for the DOE Office of Science by Argonne National Laboratory under Contract No. DE-AC02-06CH11357. E.G. is supported by NMRC/OFIRG/0032/2017 and NRF-CRP17-2017-06 grants. The authors would like to thank Drs. B.L. Roth and J. Driscoll, the National Institute of Mental Health's (NIMH) Psychoactive Drug Screening Program, for providing GPCR functional profiles. Natural Sciences and Engineering Research Council (RGPIN-2015-05939) to C.H.A.

## Author contributions

AA.H., D.I., F.L., L.B., M.S.E. designed and performed enzyme assays and biophysical experiments and analyzed data. P.L. cloned constructs. M.Sz., S.O., D.D., G.M.L., E.L.F, and Q.W. performed cell-based assays. D.I. and L.H. solved the co-structure and analyzed crystal structure. J.Bo., F.G., and J.S.Y.H. performed ChIP assays. N.P. and S.Z.A.T. performed meiosis experiments. P.J.B., R.O., M.Sc., P.K., E.G., D.B.L., C.H.A., J.M.S., S.D.K., D.J.Be., B.N., and M.V. conceived, designed, reviewed experimental data, and supervised research. B.N. and M.V. wrote the manuscript with input from all authors.

## Competing interests

R. O'Hagan, J. M. Sanders, S. D. Kattar, D. J. Bennett and B. Nicholson are current or former employees of Merck Sharp & Dohme Corp., a subsidiary of Merck & Co., Inc., Kenilworth, New Jersey, USA and may own stock or stock options in Merck & Co., Kenilworth, NJ, USA. E.G. has received research funding from Eli-Lilly and Prelude Therapeutics, has served as a consultant for Prelude Therapeutics, SK Biopharmaceuticals Korea and has served on advisory board for LION TCR and Janssen, he is a co-founder of ImmuNOA Pte. Ltd. All other authors declare no competing interests.

## Additional information

**Peer Review Information** *Nature Communications* thanks Robert Copeland and other, anonymous, reviewer(s) for their contribution to the peer review of this work. Peer reviewer reports are available.

