## [Peer Review File · Nature Communications]

Reviewers' comments:

Reviewer #1 (Remarks to the Author):

In this manuscript, Allali-Hassani et al describe the discovery and characterization of the first inhibitor of the PRDM9 histone methyltransferase.

PRDM9 is one of the 19 PRDM proteins in human. Many of these proteins are tissue-specific and are involved in differentiation regulation. While all the PRDM proteins contain a PR/SET domain, the methyltransferase activity has been reported only for some of them, including PRDM9. The H3K4 methylation by PRDM9 is responsible for positioning of meiotic recombination hotspots and several reports showed it to be involved in cancer development. The crystal structure of its catalytic domain bound to SAH and substrate H3 peptide is known.

In this study, Allali-Hassani et al performed PRDM9 inhibitor screening and SAR modification yielding a potent PRDM9 inhibitor called MRK-740. The inhibitor is specific for PRDM9. The molecular details of the MRK-740 interaction with PRDM9 were revealed by a co-crystal structure. The inhibition of ectopically expressed PRDM9 methyltransferase activity by MRK-740 in two different cell lines is shown.

The inhibitor development and biophysical and structural experiments used to characterise the affinity and specificity of its interaction with PRDM9 are elegant and well performed. The only concern I have regards the significance of this study.

1. Currently, there is no cell model for studying PRDM9 role in meiosis and the functional research thus relies on mouse models. A structure-based PRDM9 catalytic mutant mouse has been reported (Diagouraga et al, Mol Cell 2018). For analysis of the PRDM9 role in meiosis in human (or other species if MRK-740 retains its inhibition) the discovery of MRK-740 might be useful.

2. MRK-740 is highly specific for PRDM9, but the mode of interaction of its two central rings with SAM could be useful for design of inhibitors of other SAM utilising enzymes. The impact of the study would be broader if the authors could at least indicate if this approach can indeed be applied on other enzymes. Is there a similar cavity available for similar compounds in other SAM enzymes? In Prdm9, this interaction required displacement of the post-SET domain. Is that possible in some other enzymes? Can its be modelled for some other SAM enzymes to strengthen the claim of general applicability.

3. The authors also mention implication of PRDM9 in various cancers. But at the moment, PRDM9 is not a prime candidate for anti-cancer drug development. Again, the paper would be stronger if the relevance of the PRDM9 inhibitor was supported by additional experiments at least indicating some impact of MRK-740 on cancer cells. If some of the cancer cell lines, where PRDM9 is overexpressed, are available, one could analyse the impact of MRK-740 and MRK-740-NC on cancer cell proliferation?

Minor points:

1. Does the crystal structure explain why MRK-740 does not bind to PRDM7 and other PRDM proteins with known structure?
2. Figure 3. There are no residues, secondary structure nor atoms labelled.
3. The supplementary Table 1 is missing average B-factors for the refined structure.

Reviewer #2 (Remarks to the Author):

The manuscript by Allali-Hassani et al reports the first example of a potent and selective inhibitor of PRDM9. The PRDMs represent a unique branch of the protein lysine methyltransferase family that have heretofore lacked useful tool compound inhibitors. Hence, the current manuscript reports novel and important information that will be of broad interest to the readership of the journal. As such, publication is recommended once the authors have had an opportunity to address the following comments.

Major Comment

The major issue with the manuscript as currently written is that the authors only passingly mention the lack of phenotypic impact (i.e., anti proliferative activity) of the compound in cells at concentrations that should be near-saturating of target occupancy. At the same time, the authors convincingly show that the compound is cell permeable and effects inhibition of intracellular H3K4me3. The lack of phenotypic impact thus calls into question the current oncogenic hypothesis for PRDM9 activity in MCF7 and HEK293T-related cancers. This is not a failure of the inhibitor, but rather an important finding that can only be obtained with a well-designed and well-behaved inhibitor. I feel that the authors are obliged to point this out more explicitly in the Results and comment on the finding in the Discussion section.

Minor Comments: (comments below are identified by page (P) and line (L) numbering for the convenience of the authors)

P3,L10: are these mutations gain-of-function or loss-of-function? Do they contribute to the oncogenic hypothesis?

P4,L12: is the S357Y mutation representative of the cognate change from PRDM7 to PRDM9?

P4,L27: what do the authors mean by "balanced conditions". Ref 4 does not appear to be a primary reference for this concept.

P5,L8: the authors should point out that the SPR experiments were performed in the presence of saturating (?) SAM. What is the K_m for SAM? Why is the K_d obtained here so much greater than the IC_{50} reported on page 4?

P5,L25 and elsewhere: the term "dose" is used interchangeably with "concentration". This is incorrect. The term "dose" should be reserved for compound concentration or mass administered to an animal to effect a pharmacological outcome.

P6, L2-6: please provide a reference for the patterns expected for the IC_{50} as a function of [substrate] that indicate different inhibition mechanisms.

P6,L7: what is the K_i value at infinite [SAM] determined from the asymptote of the Fig 2d and at zero [peptide] from the intercept of Fig 2e? How do these two values related to one another?

P6,L25: as the compound seems to make interactions with the adenosine ring system of SAM, one would expect to see the same effect with SAH. Was this tested crystallographically or kinetically? One should see synergistic inhibition by combinations of SAH (product) and compound in Yonetani-Theorell analysis.

Signed,

Robert A. Copeland, Ph.D.

Reviewer #3 (Remarks to the Author):

Allali-Hassani and coworkers attempted to report MRK-740 as the first-in-class, potent, selective and cell active PRDM9 inhibitor. The MOA from structural studies is very interesting: SAM-uncompetitive and substrate competitive. The compound has unusual interaction with the pi system of the adenosyl group of SAM. These results are interesting and important to the drug discovery field.

Although figures 4c through 4h showed graphic diagrams of ligand binding, readers are difficult to obtain the information engendered in the figures. It would be better off that the authors add either in the main text or in the legends some concise sentences to describe key interactions between the ligands and the intermediates.

Figure 5a, for the mutant two lanes, why both are 0 uM?

Although the authors demonstrate target engagement in cells by studying H3K4 methylation, direct evidence is not presented. Can the authors use CETSA or other techniques to show the title compound binds to PRDM9 in cells?

As the authors described in the introduction part, PRDM9 is dysrelated in certain cancers. It would be highly valuable if they can test if there is any anti-cancer effect of the lead inhibitor. The reviewer understood that this could require huge work but certainly is granted in the future.

Thank you for valuable comments and suggestions.

We carefully considered all comments and performed all related experiments and revised the manuscript accordingly. We now provide direct target engagement data using NanoLuciferase thermal shift assay (NAL TSA) (new Fig. 5), data on the effect of MRK-740 on proliferation of cancer cell lines (Supplementary Fig. 13) and further investigated the biological effect of MRK-740 on meiotic progression in spermatocytes (Supplementary Fig. 14). We also revised the structural section to answer all questions, and thoroughly investigated the effect of SAH vs SAM on MRK-740 binding. We now included WaterMap simulations data to further explain the differences.

We believe the manuscript has significantly improved. Here are the Point-by-Point answers to comments:

Reviewers' comments:

Reviewer #1 (Remarks to the Author):

The inhibitor development and biophysical and structural experiments used to characterise the affinity and specificity of its interaction with PRDM9 are elegant and well performed. The only concern I have regards the significance of this study.

1. Currently, there is no cell model for studying PRDM9 role in meiosis and the functional research thus relies on mouse models. A structure-based PRDM9 catalytic mutant mouse has been reported (Diagouraga et al, Mol Cell 2018). For analysis of the PRDM9 role in meiosis in human (or other species if MRK-740 retains its inhibition) the discovery of MRK-740 might be useful.

We now included data indicating the effect of MRK-740 on spermatogenesis.

Supplementary figure 14 has been provided and the following text has been added on page 10-11:

“In order to validate the biological activity of MRK740 we investigated whether administration of this chemical probe in vivo affects meiotic integrity in male germ cells. The major phenotype of Prdm9^{-/-} mice is an arrest of meiosis during the first meiotic division resulting in infertility. Here Prdm9 is required to generate an epigenetic profile on chromatin which specifies where meiotic recombination should take place. Recombination is a prerequisite for the pairing of chromosomal homologs. As such, the loss of Prdm9 leads to failed or non-homologous pairing which arrests meiosis and leads to germ cell death 8. To bypass the difficulties of drug administration to meiotic cells posed by the blood-testis barrier, we utilized a microinjection approach where compounds were delivered directly to the testis (Supplementary Methods). To control for potential effects of drug administration we first determined that the negative control (MRK-740-NC) had no effect on meiotic progression. MRK-740-NC treated spermatocytes showed no overt meiotic defects and were able to fully pair chromosomal homologs during the pachytene substage of meiotic prophase I (Supplemental Fig. 14a). Successful pairing was determined by the co-localization of the synaptonemal complex proteins Sycp3 (magenta staining) and Sycp1 (green staining). MRK-740 treated spermatocytes however showed meiotic defects including non-homologous synapsis or failed pairing at this same stage (Supplemental Fig. 14b, d, e) and were highly similar to the defects observed in age matched Prdm9^{-/-} mice (Supplemental Fig. 14c, f, g).

2. MRK-740 is highly specific for PRDM9, but the mode of interaction of its two central rings with SAM could be useful for design of inhibitors of other SAM utilising enzymes. The impact of the study would be broader if the authors could at least indicate if this approach can indeed be applied on other enzymes. Is there a similar cavity available for similar compounds in other SAM enzymes? In Prdm9, this interaction required displacement of the post-SET domain. Is that possible in some other enzymes? Can its be modelled for some other SAM enzymes to strengthen the claim of general applicability.

- We expanded the discussion to provide further details on page 11, line 29 to page 12, line 24 (2nd paragraph of the discussion).

3. The authors also mention implication of PRDM9 in various cancers. But at the moment, PRDM9 is not a prime candidate for anti-cancer drug development. Again, the paper would be stronger if the relevance of the PRDM9 inhibitor was supported by additional experiments at least indicating some impact of MRK-740 on cancer cells. If some of the cancer cell lines, where PRDM9 is overexpressed, are available, one could analyse the impact of MRK-740 and MRK-740-NC on cancer cell proliferation?

We have tested three cell lines with detectable Prdm9 expression and three with negligible expression. The data has been presented as Supplementary figure 13 and the following text has been added on page 10:

“Effect of MRK-740 on proliferation of cancer cell lines

To determine if MRK-740 impacted proliferation of cancer cell lines, we tested three cell lines with detectable Prdm9 expression (FPKM ≥ 1) and three with negligible expression (FPKM < 1) as defined by Cancer Cell Line Encyclopedia (<https://portals.broadinstitute.org/ccle>). The cell lines that express Prdm9 included the multiple myeloma cell line SKMM2 as well as the two breast cancer cell lines CAL851 and HCC1806. The three matched control cell lines were RPMI-8226, MDA-MB-436 and HCC1143. Treatment with MRK-740 and MRK-740-NC up to 30 μ M didn't have any significant effect on proliferation of any of the cell lines tested indicating that proliferation of the Prdm9 expressing cell lines is not PRDM9 dependent (Supplementary Fig. 13).”

Minor points:

1. Does the crystal structure explain why MRK-740 does not bind to PRDM7 and other PRDM proteins with known structure?

This has been discussed in more details on page 6, line 26 to page 7 line 3 as follows:

“Significant additional interactions were observed between the adenosine moiety of SAM and the two central rings of MRK-740 (Fig. 3b). Collectively, the extensive interaction surface between SAM and MRK-740 explains why we observed SAM-dependent inhibition. None of the three residues differentiating the PRDM9 and PRDM7 PR domains (N289S, W312S and Y357S) interact directly with the inhibitor, but Y357 and N289 are within 5 and 7 Å respectively of MRK-740 and may contribute to the specificity of inhibition via second-shell factors such as the stabilization of residues directly interacting with the inhibitor.”

2. Figure 3. There are no residues, secondary structure nor atoms labelled.

- important residues are now labeled in Figure 3b

- important secondary structural elements are referred to in figure legend in color (no change)

3. The supplementary Table 1 is missing average B-factors for the refined structure.

- We added the average B-factors to the table and updated as Nature Communications template

Reviewer #2 (Remarks to the Author):

The manuscript by Allali-Hassani et al reports the first example of a potent and selective inhibitor of PRDM9. The PRDMs represent an unique branch of the protein lysine methyltransferase family that have heretofore lacked useful tool compound inhibitors. Hence, the current manuscript reports novel and important information that will be of broad interest to the readership of the journal. As such, publication is recommended once the authors have had an opportunity to address the following comments.

Major Comment

The major issue with the manuscript as currently written is that the authors only passingly mention the

lack of phenotypic impact (i.e., anti proliferative activity) of the compound in cells at concentrations that should be near-saturating of target occupancy. At the same time, the authors convincingly show that the compound is cell permeable and effects inhibition of intracellular H3K4me3. The lack of phenotypic impact thus calls into question the current oncogenic hypothesis for PRDM9 activity in MCF7 and HEK293T-related cancers. This is not a failure of the inhibitor, but rather an important finding that can only be obtained with a well-designed and well-behaved inhibitor. I feel that the authors are obliged to point this out more explicitly in the Results and comment on the finding in the Discussion section.

As mentioned above in answer to reviewer 1's comment 3, we have now tested three cell lines with detectable Prdm9 expression and three with negligible expression. The data has been presented as Supplementary figure 13. The following text has been added on page 10:

MRK-740 does not affect proliferation of tested cancer cell lines

To determine if MRK-740 impacted proliferation of cancer cell lines, we tested three cell lines with detectable *Prdm9* expression (FPKM ≥ 1) and three with negligible expression (FPKM < 1) as defined by Cancer Cell Line Encyclopedia (<https://portals.broadinstitute.org/ccle>). The cell lines that express *Prdm9* included the multiple myeloma cell line SKMM2 as well as the two breast cancer cell lines CAL851 and HCC1806. The three matched control cell lines were RPMI-8226, MDA-MB-436 and HCC1143. Treatment with MRK-740 and MRK-740-NC up to 30 μM did not have any significant effect on proliferation of any of the cell lines tested indicating that proliferation of the *Prdm9* expressing cell lines is not PRDM9 dependent (**Supplementary Fig. 13**).

Minor Comments: (comments below are identified by page (P) and line (L) numbering for the convenience of the authors)

P3,L10: are these mutations gain-of-function or loss-of-function? Do they contribute to the oncogenic hypothesis?

Stransky et.al. (ref:12) report "the Mutational Landscape of Head and Neck Squamous Cell Carcinoma" and in passing indicate that PRDM9 (11%) is recurrently mutated. However, they do not explore the consequences of such mutations.

P4,L12: is the S357Y mutation representative of the cognate change from PRDM7 to PRDM9?
Yes (PMID: 27129774; REF: 6)

P4,L27: what do the authors mean by "balanced conditions". Ref 4 does not appear to be a primary reference for this concept.

We revised the text on page 4 line 27 to clarify as: "...at concentrations of SAM and substrate equivalent to their respective Km values (balanced conditions)." Reference 4 was referred to for more information on kinetic parameters.

P5,L8: the authors should point out that the SPR experiments were performed in the presence of saturating (?) SAM. What is the Km for SAM? Why is the Kd obtained here so much greater than the IC50 reported on page 4?

SPR was originally performed in the presence of 250 μM SAM (~3.5x Km for SAM of 70 μM).

Guided by the comments, we decided to re-run the SPR at 5xKm of SAM (350 μM), which resulted in determining a much lower K_d value of 87 \pm 5 nM, much closer to the IC₅₀ value we report. We now replaced figure 2a and 2b with the new figures from the repeated experiments and IC50 value for MRK-740 with PRDM9 from 6 replicates (80 \pm 16 nM) for proper comparison.

The following has been added on page 5 line 9:

"The potency of MRK-740 was further assessed by orthogonal methods. Surface plasmon resonance (SPR) analysis also confirmed its binding to PRDM9 with a K_d value of 87 \pm 5 nM in the presence of 350

μM of SAM and k_{on} and k_{off} values of $1.2 \pm 0.1 \times 10^6 M^{-1}s^{-1}$ and $0.1 \pm 0.01 s^{-1}$, respectively, as determined by kinetic analysis of the SPR data (Figs. 2a and 2b)."

P5,L25 and elsewhere: the term "dose" is used interchangeably with "concentration". This is incorrect. The term "dose" should be reserved for compound concentration or mass administered to an animal to effect a pharmacological outcome.

On P5,L25 and elsewhere in the manuscript we replaced the word "dose" with "concentration" to correct.

P6, L2-6: please provide a reference for the patterns expected for the IC50 as a function of [substrate] that indicate different inhibition mechanisms.

We provided the following reference: Copeland, R.A. Enzymes, A Practical Introduction to Structure, Mechanism, and Data Analysis. Wiley-VCH, Inc. (2000)

P6,L7: what is the K_i value at infinite [SAM] determined from the asymptote of the Fig 2d and at zero [peptide] from the intercept of Fig 2e? How do these two values related to one another?

On page 6, L7,

The K_i value calculated from the Y intercept of Fig 2e is 65 ± 6 nM, and the K_i value estimated from asymptote of the Fig 2d was around 160 nM. The K_i value at infinite SAM concentration should reach the actual K_i value.

We now included the K_i value calculated from the Y intercept of Fig 2e on page 6.

P6,L25: as the compound seems to make interactions with the adenosine ring system of SAM, one would expect to see the same effect with SAH. Was this tested crystallographically or kinetically? One should see synergistic inhibition by combinations of SAH (product) and compound in Yonetani-Theorell analysis.

To know if SAH would have the same effect as SAM on PRDM9 inhibition by MRK-740, we performed the experiment in figure 2d at increasing concentrations of SAH and SAM in parallel (see the figure below, also provided as supplementary figure S6a). However, we didn't observe a synergistic inhibition. Curiously we also performed SPR for binding of MRK-740 in the presence of SAH in parallel with SAM (see the figure below, also provided as Supplementary figure S6b) and binding of MRK-740 to PRDM9 was dramatically weaker in the presence of SAH. Similar data obtained using ITC (data not included to avoid repetition). We reasoned that the void formed by removal of the SAM sulfonium methyl group in the hypothetical PRDM9/SAH/MRK-740 complex could trap and isolate one or more water molecules from bulk solvent, resulting in an unfavorable complex due to the inclusion of water molecules with suboptimal hydrogen bond networks. We used WaterMap simulations to analyze such an effect and concluded that the presence of the unfavorable water molecules likely explains the differential effect of SAH and SAM on inhibition of PRDM9 by MRK-740. Please see the following text which has been added to the manuscript on page 8:

"Differential effect of SAH and SAM can partly be explained by WaterMap simulation

While our structures explain the SAM dependence of MRK-740 binding, the lack of binding in the presence of SAH was not immediately obvious. We reasoned that the void formed by removal of the SAM sulfonium methyl group in the hypothetical PRDM9/SAH/MRK-740 complex could trap and isolate one or more water molecules from bulk solvent, resulting in an unfavorable complex due to the inclusion of water molecules with suboptimal hydrogen bond networks. In order to test this hypothesis, we performed WaterMap simulations^{23,24} to investigate the observation that MRK-740 displayed significantly reduced affinity for the PRDM9/SAH complex in comparison with the PRDM9/SAM complex. Indeed, three unfavorable hydration sites with relatively large and positive enthalpies (indicating unsatisfied hydrogen

bond networks relative to bulk solvent) were identified in the WaterMap simulation of the PRDM9/SAH/MRK-740 complex (red/brown spheres, **Supplementary Fig. S8a**). In the absence of MRK-740, water molecules surrounding SAH were also energetically less unfavorable, perhaps facilitating dissociation of the reaction product SAH during the catalytic cycle. In comparison, simulation of the PRDM9/SAM/MRK-740 complex did not identify any high-energy hydration sites near the location of the SAM sulfonium group (**Supplementary Fig. S8b**), suggesting that SAM is able to exclude the binding of energetically unfavorable water molecules through steric means. The exclusion of unfavorable water molecules would result in a complex of increased stability relative to the PRDM9/SAH/MRK-740 complex, consistent with our experimental observations.”

Signed,

Robert A. Copeland, Ph.D.

Reviewer #3 (Remarks to the Author):

Allali-Hassani and coworkers attempted to report MRK-740 as the first-in-class, potent, selective and cell active PRDM9 inhibitor. The MOA from structural studies is very interesting: SAM-uncompetitive and substrate competitive. The compound has unusual interaction with the pi system of the adenosyl group of SAM. These results are interesting and important to the drug discovery field.

Although figures 4c through 4h showed graphic diagrams of ligand binding, readers are difficult to obtain the information engendered in the figures. It would be better off that the authors add either in the main text or in the legends some concise sentences to describe key interactions between the ligands and the intermediates.

- We expanded the crystal structure paragraph in result section and included more information on types of interactions in 3rd paragraph in structure related section of the results on page 7-8.

Figure 5a, for the mutant two lanes, why both are 0 uM?

First and last lanes on figure 5a (now Fig. 6a) are controls (not mutant) at 0 μ M of compound for side-by-side comparison.

This has been clarified now in legend of figure 6.

Although the authors demonstrate target engagement in cells by studying H3K4 methylation, direct evidence is not presented. Can the authors use CETSA or other techniques to show the title compound binds to PRDM9 in cells?

We now provide direct target engagement data using NanoLuciferase thermal shift assay (NAL TSA) (new Fig. 5).

As the authors described in the introduction part, PRDM9 is dysrelated in certain cancers. It would be highly valuable if they can test if there is any anti-cancer effect of the lead inhibitor. The reviewer understood that this could require huge work but certainly is granted in the future.

We now added data on the effect of MRK-740 on proliferation of cancer cell lines (Supplementary Fig. 13).

We would like to thank you for reviewing our manuscript and providing excellent suggestions that significantly improved our manuscript.

Best Regards
Masoud

Masoud Vedadi, Ph.D.
Principal Investigator, Molecular Biophysics
Structural Genomics Consortium
Assistant Professor, Department of Pharmacology and Toxicology
University of Toronto
MaRS Centre, South Tower
101 College St. 7 Floor, Room 714
Toronto, ON, M5G 1L7
Tel: 416 946 0897
Cel: 416 432 1980
www.thesgc.org/biophysics

REVIEWERS' COMMENTS:

Reviewer #1 (Remarks to the Author):

In my opinion, the authors now significantly improved the manuscript and addressed all the raised points. I recommend the study for publication in Nature Communications.

Reviewer #2 (Remarks to the Author):

In this revised version of the manuscript, the authors have appropriately addressed all of the concerns and comments raised during the review of the original submission. I therefore find the current version of the manuscript to be acceptable for publication without need for further revisions.

Reviewer #3 (Remarks to the Author):

Vedadi and coworkers nicely reported a first-in-class PRDM inhibitor, MRK-740 binds to PRDM9 in a SAM-dependent manner. The data were solid and very clearly presented. The manuscript was well written. The reviewers' critiques were satisfyingly addressed. I would thus recommend acceptance of this manuscript for publication.

Page 5, line 18, citing Figs. 1c, 1d is not correct as they are not DSF data.

Resolution of Figure 7 is poor.

Though it is impressive that MRK-740 forms an extensive interaction with SAM in the binding pocket. The type of SAM-dependent inhibitors, however, likely has a disadvantage of being incapable of targeting the apoenzyme or SAH-bound form. Especially given that K_m of SAM for PRDM7/9 is great, 70 μ M, a large portion of PRDM7 in cells may exist in apo-form and so is not targeted by MRK-740.

We would like to thank the reviewers for their comments and suggestions. We believe the final version of the manuscript is much improved.

Here are the point by point response to reviewers' comments:

REVIEWERS' COMMENTS:

Reviewer #1 (Remarks to the Author):

In my opinion, the authors now significantly improved the manuscript and addressed all the raised points. I recommend the study for publication in Nature Communications.

Reviewer #2 (Remarks to the Author):

In this revised version of the manuscript, the authors have appropriately addressed all of the concerns and comments raised during the review of the original submission. I therefore find the current version of the manuscript to be acceptable for publication without need for further revisions.

Reviewer #3 (Remarks to the Author):

Vedadi and coworkers nicely reported a first-in-class PRDM inhibitor, MRK-740 binds to PRDM9 in a SAM-dependent manner. The data were solid and very clearly presented. The manuscript was well written. The reviewers' critiques were satisfyingly addressed. I would thus recommend acceptance of this manuscript for publication.

Page 5, line 18, citing Figs. 1c, 1d is not correct as they are not DSF data.

Resolution of Figure 7 is poor.

Though it is impressive that MRK-740 forms an extensive interaction with SAM in the binding pocket. The type of SAM-dependent inhibitors, however, likely has a disadvantage of being incapable of targeting the apoenzyme or SAH-bound form. Especially given that K_m of SAM for PRDM7/9 is great, 70 μM , a large portion of PRDM7 in cells may exist in apo-form and so is not targeted by MRK-740.

- Page 5, line 18, citing Figs. 1c, 1d is not correct as they are not DSF data.
 - o We now deleted citing Figs. 1c, 1d on page 5, line 18 for clarity
- Resolution of Figure 7 is poor.
 - o We now provide figure 7 in high resolution

Best Regards

Masoud

Masoud Vedadi, Ph.D.

Principal Investigator, Molecular Biophysics, Structural Genomics Consortium

Assistant Professor, Department of Pharmacology and Toxicology

University of Toronto

MaRS Centre, South Tower

101 College St. 7 Floor, Room 714

Toronto, ON, M5G 1L7

Cel: 416 432 1980

www.thesgc.org/biophysics